# Mutualism breakdown underpins evolutionary rescue in an obligate cross-feeding bacterial consortium

Ignacio J. Melero-Jiménez [1,2,3] ✉, Yael Sorokin[1], Ami Merlin[1], Jiawei Li[1], Alejandro Couce [3,4] ✉ & Jonathan Friedman [1,4] ✉

Populations facing lethal environmental change can escape extinction through rapid genetic adaptation, a process known as evolutionary rescue. Despite extensive study, evolutionary rescue is largely unexplored in mutualistic communities, where it is likely constrained by the less adaptable partner. Here, we explored empirically the likelihood, population dynamics, and genetic mechanisms underpinning evolutionary rescue in an obligate mutualism involving cross-feeding of amino acids between auxotrophic *Escherichia coli* strains. We found that over 80% of populations overcame a severe decline when exposed to two distinct types of abrupt, lethal stress. Of note, in all cases only one of the strains survived by metabolically bypassing the auxotrophy. Crucially, the mutualistic consortium exhibited greater sensitivity to both stressors than a prototrophic control strain, such that reversion to autonomy was sufficient to alleviate stress below lethal levels. This sensitivity was common across other stresses, suggesting it may be a general feature of amino acid–dependent obligate mutualisms. Our results reveal that evolutionary rescue may depend critically on the specific genetic and physiological details of the interacting partners, adding rich layers of complexity to the endeavor of predicting the fate of microbial communities facing intense environmental deterioration.

Environmental changes can lead to species becoming maladapted and decline toward extinction, but sometimes extinction can be avoided if species adapt rapidly, a phenomenon termed evolutionary rescue[1]. Much effort has been dedicated to elucidating the factors that affect the likelihood and dynamics of evolutionary rescue as it is central to several important areas including conservation, agriculture and medicine. For example, in conservation, evolutionary rescue may inform our efforts to maintain biodiversity in the face of phenomena like habitat loss or climate change[2,3]. In contrast, in the fields of medicine

and agriculture, evolutionary rescue can thwart our efforts to eliminate drug-resistant pathogens and pests[1,4,5].

Previous studies have focused on several major factors that determine the likelihood of evolutionary rescue, including the rate of environmental change (abrupt vs. gradual), dispersal rate or population size[1,4,6–9]. However, all of these studies have focused on populations composed of a single strain, whereas natural populations typically harbor multiple interacting strains and species. Interestingly, these interactions are predicted to alter the likelihood of evolutionary

[1]Institute of Environmental Sciences, The Hebrew University of Jerusalem, Rehovot, Israel. [2]Departamento de Botánica y Fisiología Vegetal, Universidad de Málaga, Campus de Teatinos s/n, 29071 Málaga, Spain. [3]Centro de Biotecnología y Genómica de Plantas (CBGP, UPM-INIA/CSIC), Universidad Politécnica de Madrid (UPM), 28223 Madrid, Spain. [4]These authors contributed equally: Alejandro Couce, Jonathan Friedman. ✉e-mail: imelero@uma.es; a.couce@upm.es; jonathan.friedman@mail.huji.ac.il

rescue[10]. Indeed, theory suggests that both competition[11,12] and cooperation[13,14] can have significant impact on evolutionary rescue dynamics. These effects are mediated by both demographic and genetic effects, as interactions can alter population sizes and introduce additional selective pressures and adaptive constraints[15,16]. Moreover, interactions can also affect adaptation to changing conditions by altering diverse population genetics parameters experienced by the interacting partners, as recently observed with a toxin from *Burkholderia cenocepacia*, which has mutagenic effects in the competing species it fails to kill[17].

An important knowledge gap is how evolutionary rescue is affected by mutualistic interactions, which play a key role in shaping communities in nature. Theoretically, evolutionary rescue of obligate mutualisms is expected to be challenging since the survival of the system is determined by the least adapted species. In this light, the weakest link hypothesis[18,19] posits that the adaptation rate is slower in mutualisms than in isolation, because while a single strain can become adapted with a single mutation, mutualistic populations require multiple mutational events to achieve the adaptation of all partners. Besides, stress can lead to a weakening of the mutualistic interactions[18], eventually leading to mutualism breakdown[20]. Examples of mutualism breakdown have been observed in natural settings, as in the interaction between corals and/or anemones and *Symbiodiniaceae* (i.e., bleaching)[21] and in the human-induced disturbance of soil microbiome[22]. Under more controlled conditions, a recent study demonstrated that obligate mutualistic *Escherichia coli* consortium have a lower likelihood of survival to increasing levels of antibiotics and that they often break down and revert to autonomy[23].

Intrigued by these results, we aimed to empirically gain further insights into how obligate mutualisms can avoid extinction when exposed to abrupt stresses, which are considered to pose the most challenging scenario for evolutionary rescue in general[7,24]. As predicted by the weakest link hypothesis, this challenge is exacerbated in obligate mutualisms, where evolutionary rescue is expected to require either rapid adaptation of both partners or, alternatively, one partner adapting to the stress while simultaneously reducing its dependence on the other. To explore this further, we exposed a synthetic two-strain *E. coli* consortium engaged in obligate cross-feeding to abrupt

environmental change imposed by two qualitatively different stressors (Fig. 1): salinity or p-nitrophenol (PNP). We chose these two treatments due to their general effect on bacterial physiology. While salinity causes hyperosmotic stress and inactivation of crucial cellular processes[25], phenolic compounds have toxic effects on cell membranes due to their high aqueous solubility[26]. Beyond quantifying the likelihood of evolutionary rescue, we also sought to characterize the evolved strains both phenotypically and genotypically in order to elucidate the mechanistic underpinnings of evolutionary rescue in our system.

Here, we show that evolutionary rescue in an obligate cross-feeding bacterial consortium is driven by the reversion of one partner to metabolic autonomy rather than by specific adaptation to the stressor. We find that most mutualistic communities exposed to abrupt lethal stress undergo evolutionary rescue; however, in all cases, only one strain survives. Genetic and phenotypic analyses show that this strain reverted to autonomy by metabolically bypassing the auxotrophy. Finally, we found that mutualistic interactions increase the population's sensitivity to multiple stressors, highlighting a general constraint of amino acid-exchanging mutualisms. Our findings suggest that the evolutionary rescue of microbial mutualistic communities may depend on the genetic and physiological traits of the partners, posing challenges for future predictions and emphasizing the need for further research on the stability of microbial mutualisms under environmental stress.

## Results

### Evolutionary rescue prevents the extinction of bacterial consortium engaged in an obligate mutualism

To test whether obligate mutualism alters the likelihood of evolutionary rescue, we evolved consortium consisting of a pair *E. coli* strains that were previously engineered to be auxotrophic to complementary amino acids[27], as well as populations of the prototrophic *E. coli* from which these auxotrophic strains were derived (see Methods). We chose knockouts in the *ilvA* (ΔI) and the *metA* gene (ΔM), because previous work has shown that they can form a robust obligate mutualism based on methionine-isoleucine exchange (Supplementary Fig. 1). Next, we propagated replicate cocultures of

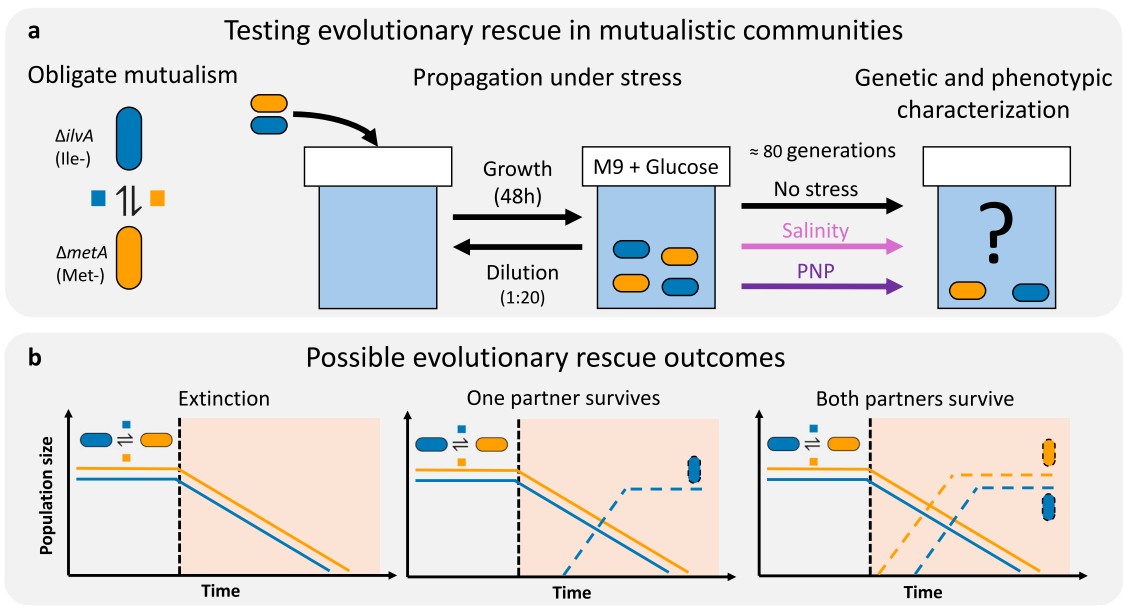

**Fig. 1 | Testing how readily evolutionary rescue can occur in populations engaged in obligate mutualism. a** We utilized a pair of *E. coli* strains engaged in obligate mutualism based on amino acid exchange and exposed them to three treatments (No stress, salinity, and PNP) to observe if the consortium could avoid extinction through evolutionary rescue. **b** Abrupt environmental stress is expected to cause a decline in population density towards extinction, followed by three possible scenarios: extinction of both partners, adaptation of one partner that recovers while the other goes extinct, or adaptation and recovery of both partners.

auxotrophic strains and monocultures of the prototrophic strain for 20 growth-dilution cycles (~80 generations) under each of three treatments ($n = 48$ for each treatment): no stress, salinity (3%), and PNP (0.4 µM).

Without stress, both mutualistic consortium and prototrophic populations maintained stable population densities (Fig. 2a). In contrast, populations engaged in obligate mutualism declined towards extinction when exposed to either stress. However, despite orders or magnitude drops in population size, the majority of mutualistic populations were able to recover and survive (37 out of 48 populations survived under salinity, and 41 out of 48 populations survived under PNP), exhibiting the typical U-shaped curve associated with evolutionary rescue[4]. No significant association between the stress conditions and survival rates was observed (Chi-squared test, $p$-value = 0.38).

We next sought to characterize whether the different treatments and replicates display common patterns or distinct dynamics of evolutionary rescue. To this end we quantified the time it took each population to start recovering, the speed of population size recovery, and the population density they finally reached (see Methods). On average, salinity-exposed populations required 2 more transfers to recover than those exposed to PNP (Fig. 2b; median of 8 $vs.$ 6 transfers; two sided Mann–Whitney $U$-test, $p$-value ~ $10^{-12}$), but once recovered they grew significantly faster (Fig. 2c; median of 0.07 $vs.$ 0.03 OD$_{600}$/Transfer; two sided Mann–Whitney $U$-test, $p$-value ~ $10^{-10}$) and to higher population densities, approaching those of the prototrophic strain (Fig. 2d; median 0.35 $vs.$ 0.19 normalized OD$_{600}$, two sided

Mann–Whitney $U$-test, $p$-value ~ $10^{-21}$). Notably, the recovery dynamics were more variable in salinity-exposed populations than in PNP-exposed ones (Levene test of equality of variances, $p$-value ~ $10^{-43}$). These findings suggest that there is a larger variability in the genetic changes underpinning evolutionary rescue under salinity than under PNP stress in our system.

We also explore the response of the auxotrophic strains when amino acids are supplemented in the medium and they do not rely on their partner (Supplementary Fig. 2a). We found that neither monoculture exhibited the population size decrease observed in the mutualistic populations. Sequencing revealed no single mutation becoming fixed in any of these populations, and those reaching high-frequency were different from the ones observed in the main experiments with the mutualists (Supplementary Fig. 2b, c). These findings support the notion that the mutualism makes populations more prone to extinction under stress.

## Stress selects for reversion to metabolic autonomy and mutualism breakdown

Next, we wanted to know if the consortium composition was altered as a result of the putative genetic changes that may underlie the observed evolutionary rescue. We tested for the presence of each strain at the end of the experiment using allele-specific PCR, looking for the positive amplification of the $ilvA$ or $metA$ genes, respectively (see Methods, Supplementary Table 1). We first confirmed that both strains coexisted until the end of the experiments under stress-free conditions. However, only the ΔI strain survived in all end-point

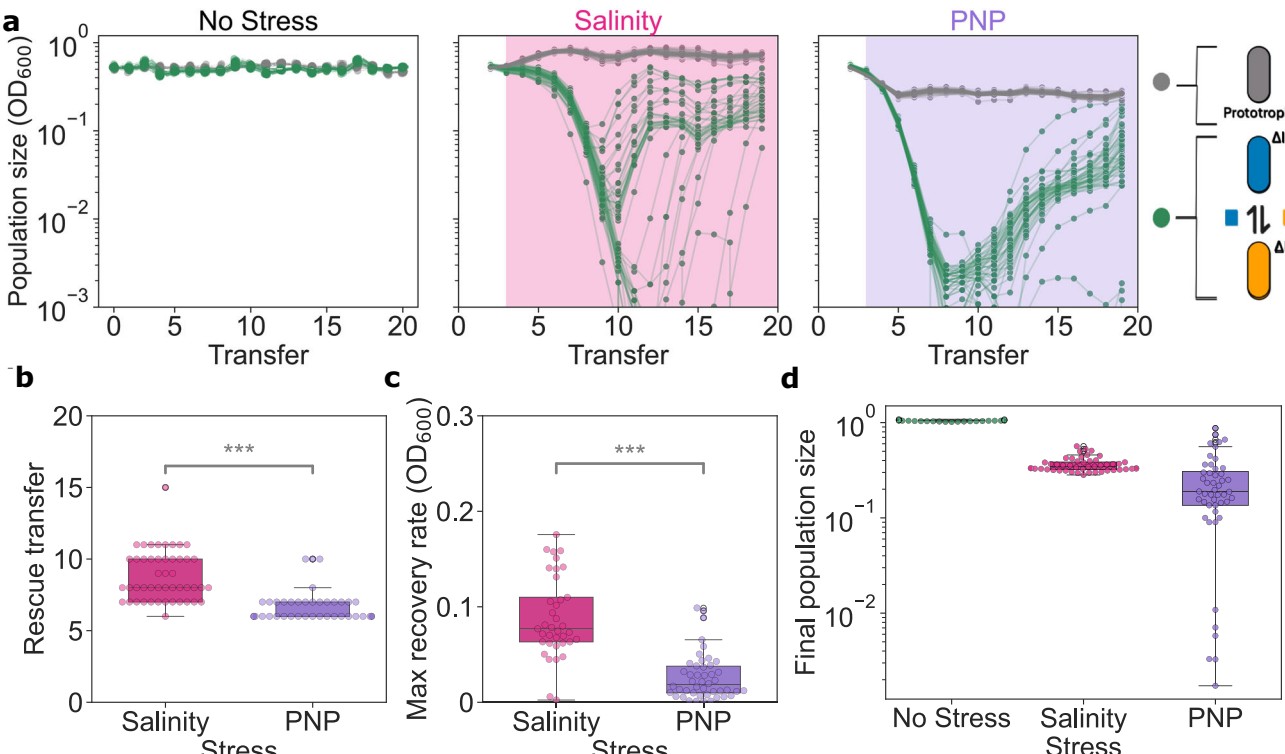

**Fig. 2 | Evolutionary rescue prevents the extinction of bacterial consortium engaged in an obligate mutualism based on metabolic exchange. a** Population dynamics of the prototrophic strain (gray) and the obligate mutualism (green) in three different stress treatments (no stress, salinity, and PNP). The red and purple background indicate exposure to salinity or PNP, correspondingly. The experiment consisted of 48 independent populations for each treatment. **b** Box plots representing the transfer when consortia/populations begin to recover after stress exposure (OD$_{600}$ shifts from negative to positive trend), with $n = 48$ for salinity and PNP. $P$-value (***$p < 0.01$) was determined by a two-sided Mann–Whitney $U$-test ($p$-value ~ $10^{-12}$). **c** Maximal rate of recovery computed as maximum change in

population size per transfer for each experimental culture after stress exposure (OD$_{600}$/transfer), with $n = 37$ for salinity and $n = 46$ PNP. $P$-value (***$p < 0.01$) was determined by a two-sided Mann–Whitney $U$-test ($p$-value ~ $10^{-10}$). **d** Final population size of mutualistic communities ($n = 48$), calculated as the median of the last three transfers normalized relative to the median of the prototrophic strain at the final transfer. $P$-values (**b, c**) ***$< 0.001$ (two sided Mann–Whitney $U$-test). All box plots display the interquartile range (IQR) of the data, with the horizontal line inside the box indicating the median. The whiskers extend to 1.5 times the IQR, showing the range of the data distribution. Source data are provided as a Source Data file.

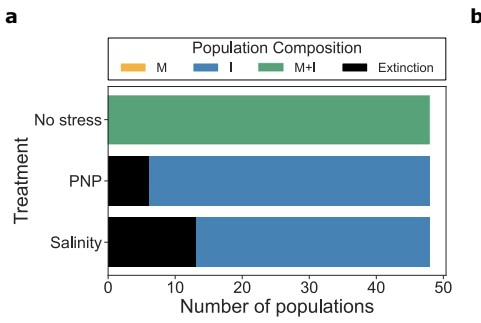

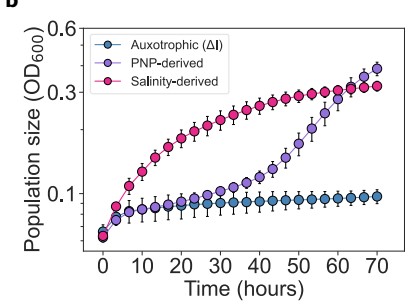

**Fig. 3 | Only a single strain survives the stress and reverts to metabolic autonomy. a** Strains present at the end of the experiment. Colors indicate the presence of each strain in the consortium as detected by PCR. **b** Growth curves of recovered populations and of the auxotrophic ancestor of the ΔI strain in M9 without isoleucine addition. Different colors indicate the treatment from which the populations were isolated (purple from PNP, red from salinity and blue is the

ancestor ΔI strain). Dots and error bars indicate the mean ± SD of measurements from three technical replicates of each of three evolutionary replicates (i.e. three different populations that were evolved in parallel during the evolutionary rescue experiment). Growth curves of individual replicates are included in Supplementary Fig. 3. Source data are provided as a Source Data file.

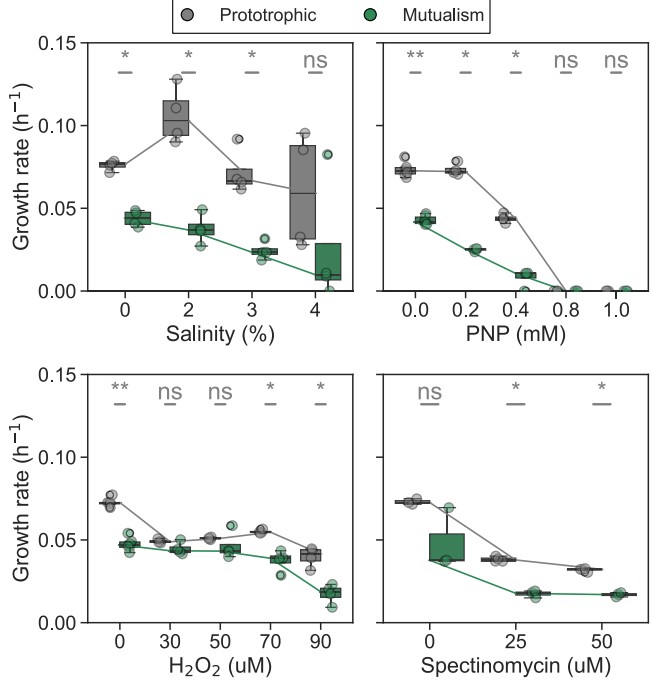

**Fig. 4 | The obligate mutualism is more susceptible to environmental stress than the prototroph.** Each panel shows the growth rates of prototrophic (gray) and mutualistic consortium (green) under different stressors: salinity (%), p-nitrophenol (PNP, mM), hydrogen peroxide ($H_2O_2$, μM), and spectinomycin (μM). Each box plot displays the interquartile range (IQR) of the data, with the horizontal line inside the box indicating the median. The whiskers extend to 1.5 times the IQR, showing the range of the data distribution (n = 4 for all boxplots except PNP (0) and $H_2O_2$ (0), where n = 6, and spectinomycin (0), where n = 3). The lines connecting the boxplots in each panel indicate the median growth rates for the prototrophic and mutualistic groups across different conditions. *P*-values (*p < 0.05, **p < 0.01, ns not significant) were determined by a two-sided Mann–Whitney *U*-test. Source data are provided as a Source Data file.

populations subjected to either stress (Fig. 3a). The ancestral strains lacked known phages or plasmids that may mediate the horizontal transfer of genes between the two auxotrophs, so the absence of the *ilvA* gene in our populations suggests that evolutionary rescue was achieved by rapid reversion to metabolic autonomy. To confirm this possibility, we cultured these populations in M9 liquid media lacking isoleucine. All populations grew without any supplemented isoleucine (Fig. 3b), indicating the acquisition of some

compensatory mutations that lead to the reversion from auxotrophic to prototrophic states.

## Mutualists are less tolerant to high-stress levels compared to the prototrophic strain

Since the prototrophic strain was less affected by either salinity or PNP and did not decline toward extinction during the experimental evolution (Fig. 2a), we hypothesized it is less sensitive to stress than the mutualism. To test this hypothesis, we exposed monocultures of the prototroph and cocultures of auxotrophic strains to several stresses with different modes of action: salinity, PNP, hydrogen peroxide, and the antibiotic spectinomycin. The mutualism grew slower than the prototroph under non-stress conditions (Fig. 4, mean exponential growth rate of 0.07 ± 0.01 *vs.* 0.04 ± 0.01 ($h^{-1}$); two sided Mann–Whitney *U*-test, *p*-value ~ $10^{-7}$). However, they grew by a similar amount after 48 h (Supplementary Fig. 4, mean number of doublings 3.1 ± 0.01 *vs.* 2.8 ± 0.6; two sided Mann–Whitney *U*-test, *p*-value = 0.07). Growth rates and yields declined as stress levels increased, with the mutualism consistently showing lower growth rates and yields than the prototroph across all four types of stress (Fig. 4, Supplementary Figs. 5 and 6). Our results are consistent with previous experiments that reported heightened sensitivity of mutualistic interactions to stress conditions[23,28,29] and indicate that the prototrophic strain is able to survive stress levels that would drive the mutualism toward extinction.

## Mutations in genes involved in amino acid biosynthesis drive the evolutionary rescue

Since the prototrophic populations were less sensitive to either salinity or PNP (Fig. 4), we hypothesized that evolutionary rescue in our system was due to bypassing the auxotrophy, rather than stress-specific adaptation. To test this hypothesis, we quantified the growth of strains isolated from the recovered populations when exposed to each of the individual stressors.

We observed that the growth rate of the rescue strains was higher in almost all the cases than that of the mutualism when exposed to either stress (Fig. 5a). Furthermore, both the growth rate and the yield of the rescue strains was similar regardless of their evolutionary history when exposed to either condition (Supplementary Fig. 7), indicating a substantial degree of cross-resistance despite the marked differences in the nature of both stresses (osmotic imbalance *vs.* membrane disruption). The fact that the strains recovered from one stress did not outperform those evolved under the alternative stress lends support to our hypothesis that evolutionary rescue was driven mostly by bypassing the auxotrophy.

To identify the mutations and putative targets of selection underpinning the observed evolutionary rescue, we subjected the

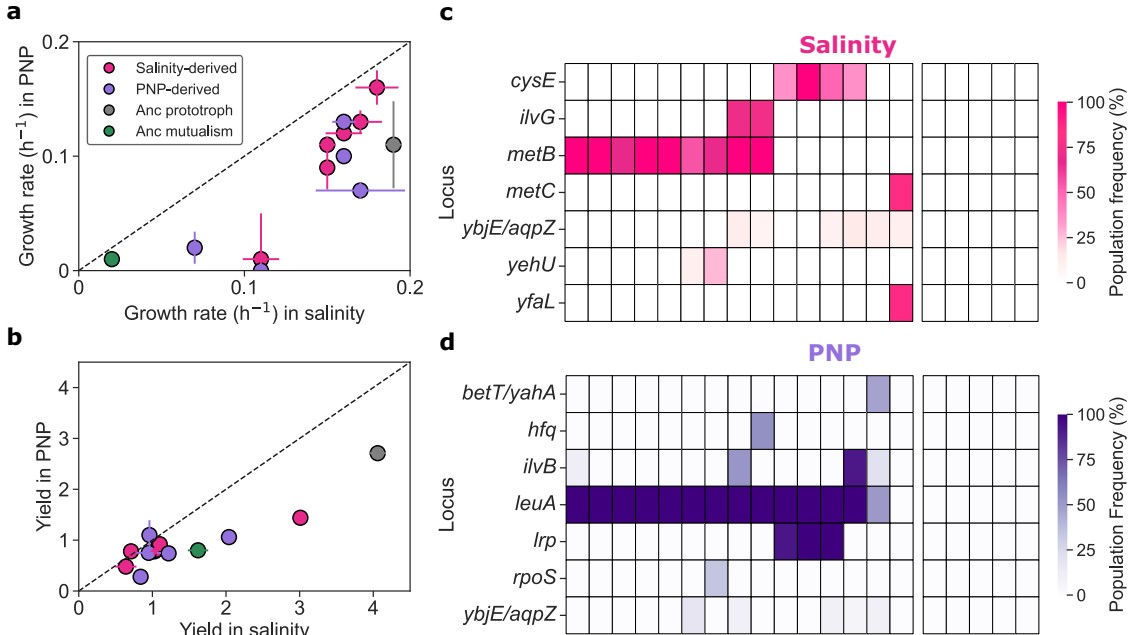

**Fig. 5 | Metabolic pathway mutations drive evolutionary rescue and mutualism breakdown. a** Growth rates and (**b**) yields of the mutualism, the prototrophic ancestor, and strains evolved in salinity or PNP. For this experiment, we included one population of the prototrophic strain (gray), six evolutionary replicates derived from salinity stress and five from PNP stress, and the ancestral M and I strains that constituted the mutualism. The data are presented as the mean ± SEM ($n = 4$).

**c**, **d** Genes with mutations present in at least two replicate populations or occurring at a frequency > 0.3, represented by different colors (red for salinity and purple for PNP). Color shades indicate the frequency of each mutation, with darker shades indicating higher mutation frequencies. Specific mutations are detailed in Supplementary Tables 2 and 3. Source data are provided as a Source Data file.

recovered populations to whole-population, whole-genome sequencing. We identified a total of 51 mutational events (22 and 29 under salinity and PNP, respectively) of various types within the derived strain collection (Supplementary Table 2 and 3). The most common alterations were non-synonymous single-nucleotide polymorphisms (SNPs), which made up 80% of the total mutations. Specifically, GC → AT and AT → GC transitions were particularly prevalent, accounting for 29% and 31% of all base substitutions, respectively (Supplementary Fig. 8). This aligns well with previous reports of their predominance in bacterial mutation patterns[30], suggesting that the environmental stresses we used did not introduce significant biases in the overall mutation rates[31,32].

Moving on to the presumed targets of adaptation, our results show that most high-frequency mutations are related to amino acid metabolism, further supporting the idea that the primary driver of the rescue was the reversion to autonomy rather than stress-specific adaptation. We identified several mutations in metabolic genes that appear to compensate for the loss of *ilvA* gene function in the salinity-evolved populations (Fig. 5c, Supplementary Table 2). In most cases, a putative metabolic mechanism underlying the compensatory phenotype can be identified. The most direct example occurs in the case of *metB*, the most prevalent target of mutation detected. MetB normally produces L-cystathionine and succinate from L-cysteine and O-succinylhomoserine in the methionine biosynthesis pathway. However, when the levels of the substrate L-cystein are low, this enzyme displays a secondary activity, catalyzing the generation of 2-oxobutanoate[33]. This compound is precisely the main product of IlvA, and represents the first step in the synthesis of isoleucine. It is therefore reasonable to speculate that the mutants of MetB we detected could potentially enhance the secondary activity of the enzyme and generate sufficient 2-oxobutanoate so as to bypass the blockade of the first step in isoleucine biosynthesis of the Δ*ilvA* background[34].

A related argument can be proposed for the second most prevalent target of mutation, *cysE*. This enzyme carries out the first step in the pathway of cysteine biosynthesis, so a putative mechanism is that *cysE* mutants result in lower concentration levels of L-cystein, which are the conditions in which wild-type MetB favors the production of 2-oxobutanoate. Along this line, we detected one case that seems explained by mutations in MetC. This enzyme normally catalyzes the reaction step that follows the action of MetB in the methionine biosynthetic pathway. However, this enzyme can also exhibit a secondary activity, converting L-cysteine to 2-aminoacrylate and hydrogen sulfide[35]. It is tempting to hypothesize that the MetC mutant observed favors this reaction, reducing overall L-cystein levels and therefore potentiating the generation of 2-oxobutanoate by MetB. Finally, one further evidence in favor of these explanations is that these three genes are present in 14/15 of the sequenced populations, and most importantly, when mutations are found in one of them, none are found in the others, suggesting that these mutations indeed mediate redundant ways to bypass the isoleucine auxotrophy. Notably, none of these mutations were found in the monocultures when they were supplemented with their corresponding amino acid and exposed to salinity (Supplementary Fig. 2).

In the PNP treatment, 14 out of 15 sequenced populations had mutations in *leuA* gene (Fig. 5c, Supplementary Table 3), which catalyzes the first step in leucine biosynthesis. The enzyme LeuA is known to have promiscuous activity toward alternative substrates 2-ketobutyrate[36] and (S)-2-keto-3-methylvalerate[37], the first and last step in the isoleucine biosynthesis pathway. A tantalizing possibility is that the mutations detected may make the reverse reactions more likely to occur for a given concentrations of substrates and products, as production of either 2-ketobutyrate or (S)-2-keto-3-methylvalerate would suffice to bypass the blockade caused by the *ilvA* knockout[38–41].

To confirm that mutations in metabolic genes were responsible for the bypass of isoleucine auxotrophy, we leveraged the fact that many salinity- and PNP-stress-derived populations showed only single

mutations in these genes. We thus isolated colonies from these populations and sequenced them to verify no other mutations were present in their genomes (Fig. 5). The isolates varied in their degree of compensation for isoleucine auxotrophy, even when mutations occurred within the same gene (Supplementary Fig. 9). For example, the *T322M* mutation in *metB* almost fully restored growth to proto-trophic levels, while the *N15S* mutation provided weaker compensation. Mutations in *leuA*, overall, resulted in a lower degree of compensation compared to those in *metB*.

## Discussion

Since we exposed an obligate mutualism to abrupt stress, we expected that evolutionary rescue would be (1) rare and (2) mediated by two different mechanisms: either stress adaptation of both strains, or one strain adapting both to the stress and becoming able to grow independently (Fig. 1). We found that evolutionary rescue occurred frequently in our system, likely in large part due to the fact that reverting to metabolic autonomy appears to suffice to also overcome the salinity or PNP stress. We speculate here that strains in mutualism are particularly sensitive to stress because they face two simultaneous challenges: coping with stress while maintaining mutualistic interactions to acquire essential amino acids. Unlike prototrophic strains, auxotrophs rely entirely on metabolite exchange, which may involve increased membrane permeability, making them more vulnerable to stressors like osmotic changes or toxins. Alternatively, stress may reduce amino acid biosynthesis due to resource reallocation, impairing protein synthesis, slowing consortium growth, and weakening its response to environmental challenges.

This latter possibility has been suggested in legume-rhizobia mutualisms, where bacteria often stop fixing nitrogen in nitrogen-rich soils, conserving energy that would otherwise support the plant and redirecting it toward their own growth[42]. Similarly, in coral-algae mutualism, algae may reduce the nutrients they share with their coral hosts when nutrients are abundant[43]. However, this hypothesis is inconsistent with our observation that the salinity sensitivity of the ΔI strain was not relieved by isoleucine supplementation (Supplementary Figs. 10 and 11). Moreover, while the rescued populations showed reduced sensitivity to both stresses (Fig. 5a, b), the specific mutations that underlie their recovery did depend on the stress in which they evolved. These results suggest a complex interplay between cellular metabolism and stress adaptation, similar to those recently described for the evolution of aminoglycoside resistance[44].

Notably, in all recovered populations, it was always the ΔI strain that avoided extinction. We found no evidence that the ΔM strain has a heightened stress sensitivity that may lead to rapid extinction before generating adaptive mutations (Supplementary Fig. 2). Instead, we found that ΔI outnumbered ΔM at equilibrium under benign conditions by a ratio of 2.5:1 (Supplementary Table 4), this difference likely provides ΔI with a larger effective mutation supply, increasing its chances to evolve and recover compared to ΔM. This asymmetry may stem from methionine's critical role in initiating protein synthesis, which could limit the ΔM auxotroph's growth when methionine is scarce. Moreover, besides population size, we cannot rule out that ΔI has a larger mutational target size for compensatory mutations compared to ΔM[45], as suggested by the diversity of recovered mutations and loci linked to amino acid biosynthesis (Supplementary Tables 2 and 3). More broadly, these observations illustrate that mutualisms are most likely inherently asymmetric: strains (let alone species) engaged in mutualism are unlikely to have identical equilibrium ratios, mutation rates, or stress sensitivities; and these asymmetries are probably a major destabilizing factor in nature that warrant future theoretical and empirical investigation.

While evolutionary rescue via stress-induced mutualism breakdown and extinction of one of the partners occurred in our system, different outcomes are likely in other settings. Mutualism breakdown may be less frequent when reverting to autonomy does not result in lower stress sensitivity, as evolutionary rescue would then require two different adaptations. Even when reverting to autonomy leads to stress adaptation, the outcome of stress exposure would depend on the rates with which the mutualistic partners can do so. In our system, a mismatch between the rates at which the strains could bypass their auxotrophies likely contributed to the breakdown of the mutualism. A lower rate of auxotrophy bypass would likely have led to fewer populations recovering. In contrast, if both the ΔI and ΔM strains could readily bypass their auxotrophies, recovery rates would be higher but the outcome more varied—rather than the only the ΔI strain recovering, either one of the strain or both of them could survive.

Here, we selected stressors that induce specific cellular stresses, challenging the survival of both partners. However, the breakdown of mutualism observed here may also occur in other mutualisms and environmental stresses. Our results are particularly relevant for understanding when breakdowns can occur in other bacterial mutualisms, such as cross-protection or nutrient exchange. The breakdown may also happen in other environments where engaging in a mutualism constrains the adaptation of both partners, thus creating a situation in which becoming autonomous may be the most effective strategy for facing stress. For example, in nutrient-limiting conditions, the growth of both partners would be reduced, and they would already find it hard to exchange those amino acids, which likely ends in the extinction or breakdown of their mutualistic interaction. This scenario mirrors what happens in nature in fungi-plant mutualisms[46]. Under nutrient deficiencies, plants cannot provide sufficient amount of carbohydrates to fungi; they further weaken the mutualism and generally lead to declines in health of mutualism[47,48]. It has been also observed that nitrogen-fixing bacteria disrupt the mutualism between legume and rhizobia under nutrient-poor soils[42,49]. Other stresses, such as thermal stress, can disrupt metabolic processes essential for amino acid exchange by impairing enzyme activity and membrane stability, which can lead to reduced nutrient transport and potential mutualism breakdown. We hypothesize that thermal stress will hinder the transport of amino acids in our system in a similar way that increases in ocean temperature disrupt nutrient exchange between coral and their symbiotic algae to cause coral bleaching[50].

Understanding the processes that lead to the breakdown of mutualistic interactions could be useful in many fields. In agriculture, identifying when plant-microbe mutualism[47] as those between nitrogen-fixing bacteria and legumes[51] begin to break can inform soil management and the selection of resilient bacterial strains, thereby reducing crop loss[46,52]. In medicine, insights into the thresholds at which host-microbe mutualisms[53,54] break—whether due to antibiotics[55], dietary changes[56], or infections[57]—could guide preventive therapies using targeted prebiotics or probiotics to maintain a balanced microbiome[58,59]. In biological conservation, the capacity to anticipate mutualism breakdown allows design more effective conversation measures and politics.

Further research is needed in order to gain a broader understanding of the factors affecting the likelihood and dynamics of evolutionary rescue of mutualisms. Here, we focused on a mutualism based on amino acid exchange between two strains of the same species. The extent to which our findings extend to other mechanisms of co-dependence, involving more distinct species, and in more natural settings remain unclear. If stress is conducive to mutualism breakdown, how come mutualisms are so prevalent in nature? Future work in other model systems will help elucidate the generality of our results. Nonetheless, real-word instances of stress-induced mutualism breakdown have been observed, including breakdown of cooperation in host-microbiota associations exposed to antibiotic treatments[60], as well as in coral bleaching events[21]. Given the prevalent environmental changes instigated by anthropogenic activities, it is crucial that we gain a deeper understanding of how mutualistic and other ecological

interactions affect a community's adaptive potential and long-term persistence in changing environments.

# Methods

## Strains

*Escherichia coli* used in this study were based on the EcNR1 *E. coli* derivative of MG1655. We used the auxotrophic strains ΔmetA (hereafter referred to as 'ΔM'), ΔilvA (hereafter referred to as 'ΔI') and the WT (hereafter referred to as prototrophic). The amino acid auxotrophs were generated by Red-recombineering with a chloramphenicol resistance cassette. Information and strains were obtained from Mee et al.[27]. Before starting the experiments, we verified the absence of the *ilvA* and *metA* genes by the lack of amplification using specific primers, detailed in Supplementary Table 1. We chose these strains because they have been previously demonstrated to reciprocally exchange amino acids and engage in obligate mutualism.

## Culture conditions

Evolutionary experiments were performed in 96 deep-well plates (maximal volume: 1 ml, Thermo Scientific Nunc) with M9-glucose media [1X M9 salts supplemented with 2 mM MgSO₄ · 7H₂O, 0. 1 mM CaCl₂, 1X trace metal solution (Teknova), 0.083 nM thiamine, 0.25 µg/L D-biotin, and 1% (wt/vol) glucose] at 30 °C and were shaken at 900 rpm for 48 h. Auxotrophic monocultures were supplemented with 100 µM of either isoleucine or methionine.

For plating, we used M9 agar plates (5 g/L peptone BD difco, BD Bioscience; 3 g/L yeast extract BD difco, BD Bioscience, 15 g/L agar Bacto, BD Bioscience). To differentiate between both strain we use selective plates with the correspond amino acid (100 µM). Plates were incubated at 30 °C for 48 h and colonies were counted manually.

## Population size and growth rate measurements

To determine the growth rate of all populations, growth kinetic experiments were performed with monocultures of the prototrophic strain and cocultures of both auxotrophic strains (ΔI and ΔM). Growth experiments were performed in 96 well plates (200 ul). The optical density was measured in two automated plate readers simultaneously, Epoch2 microplate reader (BioTek) and Synergy microplate reader (BioTek), and was recorded using Gen5 v3.09 software (BioTek). Plates were incubated at 30 °C with a 1 °C gradient to avoid condensation on the lid and were shaken at 250 rpm. OD₆₀₀ was measured every 5 min. Each strain was measured in four technical replicates.

Before calculating the yield and growth rate, a smoothing process was applied to the data. This involved averaging measurements taken every 5 min over 8 data points. The yield was computed as the logarithm of the ratio between the final OD₆₀₀ (ODf) and the initial OD₆₀₀ (ODi) concentration:

$$Yield = \log_2\left(\frac{OD_f}{OD_i}\right) \tag{1}$$

The Growth rate (h⁻¹) was calculated as:

$$Growth\ rate\left(h^{-1}\right) = \frac{1}{t_{thr}}\log\left(\frac{threshold}{OD_i}\right) \tag{2}$$

where a threshold (*threshold*) was defined as 1.5 times the initial OD concentration.

## Toxicity assays

To determine the toxicity of several environmental stresses, growth kinetic experiments were performed, as described before, with monocultures of the prototrophic, ΔI and ΔM as well as cocultures of both strains. The assay was performed in in a 96-well plate which was filled with fresh M9 with or without amino acids and supplemented with increasing amounts of salinity (0, 1, 2, 3, 4% NaCl), PNP: p-nitrophenol (0, 0.2, 0.4, 0.8, 1 mM), hydrogen peroxide (0, 30, 50, 70, 90 H₂O₂), Spectinomycin (0, 25, 50 ug/mL) and inoculated with ∼10⁵ bacteria per well. The plate was incubated for 48 h at 30 °C with shaking.

## Evolutionary rescue experiment

During the evolution experiment, monocultures of the prototrophic and isoleucine- and methionine-auxotrophic strains were grown without any externally supplied amino acid. All the strains involved in the evolution experiment started from one cryogenic stock. Before of the experiment, strains were first picked from an overnight colony into LB–Lennox medium (10 g/L bacto tryptone, 5 g/L NaCl, 5 g/L yeast extract) with chloramphenicol (20 µg/mL, Sigma cat# C0378). After 24 h, late–exponential-phase cells were harvested and washed twice in M9 salts (6 g/L Na₂HPO₄, 3 g/L KH₂P₄, 1 g/L NH₄Cl, 0.5 g/L NaCl). Then, all cell concentrations were adjusted to 0.1 OD₆₀₀ using M9 media. Coculture growth was performed by equal-volume inoculation of each strain at a seeding OD₆₀₀ of 0.05. Precultures were then used to inoculate 46 replicates for each of the two cultures: (1) monoculture of prototrophic and (2) coculture of ΔI and ΔM. In the case of both auxotrophic monocultures, precultures were used to inoculate 96 replicates in M9 media containing both amino acids (100 µM each). Cultures were grown in a volume of 800 µl in 96-well plates at 30 °C and were shaken at 900 rpm for 48 h. 40 µl of the resulting cultures were transferred every 48 into 760 µl of fresh medium. During the first three growth cycles, no stress treatment was applied to the cultures to allow populations to equilibrate. At the fourth transfer, each test culture was split up into three different environments: control (no stress), salinity (3% NaCl) and PNP (0.4 mM). Growth of all cultures was tracked by quantifying their population density (OD₆₀₀), while propagating them to fresh medium with the specific stress.

These two stresses were chosen to maximize differences in terms of mode of action, and are relevant pollutants in the environment. Salinity exerts its influence primarily through osmotic stress, where high salt concentrations challenge microbial cells by disrupting their osmotic balance. On the other hand, PNP acts as a direct chemical stressor by interfering with critical cellular processes, including enzyme functions.

## Dynamics of the U-shaped evolutionary rescue curve

Before performing the any calculation, we smooth the data by applying a moving average smoothing technique with a window size of three transfers.

The *Rescue Transfer* is defined as the transfer when an experimental culture exhibits the first increase in population size (OD₆₀₀) after being exposed to environmental change, either salinity or the presence of PNP.

The *Maximum recovery rate* is defined as the maximal positive change in population size (OD₆₀₀) per transfer for each experimental culture after the exposure to stress.

## Sequencing and genomic analyses of rescue populations

Each ∼20 generations all populations were frozen at −80 °C with 50% glycerol in a 96-deep well plate. In order to do population sequence, we use a loop and resuspend in 3 mL LB at 30 °C, overnight with shaking. We extracted genomic DNA using the Norgen Biotek Corp Bacterial Genomic DNA Isolation Kit (Cat No. 17900) and sent it to the sequencing facility at SEQCENTER (PA, USA, https://www.seqcenter. com/). The samples were sequenced on an Illumina NextSeq 2000. Demultiplexing, quality control, and adapter trimming were performed by the sequencing center with bcl-convert. Average coverage for short reads is 121 with a standard deviation of 40. Analysis of the NGS sequencing data from samples was performed using breseq (v.0.37) software tool[61,62]. This tool is free-access, based on Linux, and

has been extensively used to analyze genome's sequencing data. To prepare the input for breseq, we first extract fastq.gz files containing raw sample genomic data into fastq files. These files were then used for the analysis. In addition, we used the *Escherichia coli* str. K-12 substr. MG1655, complete genome genebank (.gbk) extension file as the reference sequence, since it has already been annotated. Additionally, we remove possible mutations due to our ancestor by comparing the complete genome sequence with the sequence of our ancestor. We run breseq in a population default mode without any modifications.

### Reporting summary

Further information on research design is available in the Nature Portfolio Reporting Summary linked to this article.

## Data availability

The full dataset used in this study is available on GitHub at https://github.com/ignamel/ER_mutualism.git and on Zenodo at https://doi.org/10.5281/zenodo.14992670[63]. All raw sequencing data is available on the US National Center for Biotechnology Information (NCBI) Sequence Read Archive (SRA) under BioProject PRJNA1143481 (http://www.ncbi.nlm.nih.gov/bioproject/1143481). Source data are provided as a Source Data file. Source data are provided with this paper.

## Code availability

The Python 3.11.9 code used for analyzing, and creating figures is open-source and available on GitHub (https://github.com/ignamel/ER_mutualism.git) and stored at https://doi.org/10.5281/zenodo.14992670[63].

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

## Acknowledgements

We thank Valeria Tsvichenko and Rotem Shner for the laboratory assistance, and members of the Friedman lab and Couce lab for helpful discussions. JF was supported by the Israel Science Foundation (grant No. 883/22). AC supported by the Agencia Estatal de Investigación (Centros de Excelencia "Severo Ochoa", SEV-2016-0672 and CEX2020-000999-S; Proyectos de I+D+i, PID2022-142857NB-I00), and a Comunidad de Madrid "Talento" Fellowship (2019-T1/BIO–12882, 2023-5 A/BIO-28940). IJMJ was supported by "Margarita Salas" post-doctoral Fellowship (MS2021_003, Universidad de Málaga, Unión Europea–NextGeneration EU, Ministerio de Universidades, Spain).

## Author contributions

I.J.M.J., A.C. and J.F. designed the study. I.J.M.J., Y.S., A.M. and J.L. performed the experiments. I.J.M.J, A.C. and J.F. performed the analysis. I.J.M.J, A.C. and J.F. wrote the manuscript.

## Competing interests

The authors declared no competing interests.
