## [Transparent Peer Review file · Nature Communications]

Mutualism breakdown underpins evolutionary rescue in an obligate cross-feeding bacterial consortium

Corresponding Author: Dr Ignacio Melero-Jiménez

Version 0:

Reviewer comments:

Reviewer #1

(Remarks to the Author)

Overview:

This paper explores how an obligate mutualism consisting of two engineered *E. coli* strain respond to environmental stress. One strain is auxotrophic for the amino acid methionine (ΔmetA) and the other strain is auxotrophic for the amino acid isoleucine (ΔilvA). These strains were passaged in 96 well plates for the equivalent 80 generations under three conditions: no stress, salinity stress, or p-nitrophenol stress. Although a few populations went extinct, most survived. When exposed to stress, this survival always involved the loss of the ΔmetA . The persistence of ΔilvA was attributed to a “bypassing of the auxotrophy”. Based on whole genome sequencing of isolates, different mutations arose under different stress treatments, which may have allowed ΔilvA to revert to an autonomous lifestyle. These findings are discussed in the context of evolutionary rescue and address questions about mutualism breakdown. The paper is reasonably well written. The figures are clear and to the point.

Major concerns:

It is curious that, from a demographic and evolutionary perspective, the mutualism seems very asymmetric. The ΔmetA strain is always lost, but we don't really learn much about why that's the case. Is the reason for the loss of ΔmetA due to its critical role in initiating protein synthesis, its involvement in key metabolic pathways, or its contribution to the bacterial stress response and adaptation? From Fig. 2A, it is difficult to assess the dynamics of the individual strains. It's possible that ΔmetA rapidly goes to extinction, perhaps in just a transfer or two. Might this mean that the outcomes were inevitable with the ΔmetA - ΔilvA pairing? Was the deck stacked against ΔmetA from the very start? If so, perhaps the findings are not very generalizable or as interesting as the paper suggests?

The paper provides three explanations for how ΔilvA may have evolved autonomy during the experiment. These seem like plausible and testable hypotheses. Given the tractability of the model organism (*E. coli*) it's not totally unreasonable to expect that some of these mutations could have been introduced into “clean” auxotrophs to see if they could recapitulate the phenotype of the evolved mutant. Such findings would support the assertion that these mutations are in fact compensatory, and would allow for deeper insight into the mechanisms of defection.

Some additional experiments came to mind that might assist with understanding the population dynamics. It might be interesting to see if similar mutations would arise if ΔilvA strains were grown over 20 transfers in media supplemented with isoleucine and exposed to stress. If ΔmetA strains were grown with supplemented methionine or mutualist-derived media and grown under the same conditions, would they still go extinct? If findings from single-strain experiments are different from cocultures, then I believe that would make a strong argument for the mutualism affecting evolutionary rescue, as stated in the introduction.

I appreciate the use of a model system to test general ideas, so perhaps this is somewhat semantic, but an ecological “community” is made up of two or more species. Similarly, mutualisms involve two or more species. In this case, the experiments involve two engineered derivatives of MG165 *E. coli*. In other words, this is very much a population, not a community.

Other concerns:

The section titled "Mutualists can survive lower stress levels than the prototrophic strain" seems a bit out of place. The title is confusing; it reads as though the mutualists have an ability that the prototrophs do not, when the mutualists have an inability to survive the same caliber of stress as the prototroph.

In the last sentence of the section mentioned above, it seems there is an error reading, "the auxotrophic strain is able to survive stress levels that would drive the mutualism towards extinction." Is this supposed to be "prototrophic"?

"Toxicity assays" has a typo in line 4 ("ot" instead of "or").

(Remarks on code availability)

Reviewer #2

(Remarks to the Author)

This manuscript by Meleró-Jiménez and colleagues explores the dynamics of evolutionary rescue in a microbial community that is designed to have an obligate mutualism through the cross-feeding for amino acids. The authors provide strong evidence that more than 80% of communities avoid extinction under stress conditions, and that the primary mechanism for this the rescue is reversion to metabolic autonomy in one of the partners. These results have significant implications for the understanding of microbial community resilience under environmental stress, especially in the context of mutualistic relationships. I think the manuscript is well written, methodologically sound, and combines ecological data with genomic data provides interesting insights into the role of interactions and mutations in evolutionary rescue. The writing is very clear and I appreciate the hypothesis driven design of the work

However, I think several points need further clarification and expansion to strengthen the conclusions drawn and address potential gaps.

1. Title: I think the title could be made more specific- What is missing is that reversion of metabolic autonomy causes a breakdown in the mutualism (in most cases). The current title leads one to believe that the evolutionary rescue saved the mutualism.
2. Discussion: I would have liked a more detailed discussion of what this work can inform one about the stability of microbial/mutualistic communities and where it could be useful. The authors allude to this in in the start of the main text (lines 53-56) but never revisit this again.
3. Can the authors provide (reasonable) speculation as to what mechanisms (molecular or otherwise) that strains that constitute mutualistic interactions are sensitive to stress?
4. Since the reversion is driven in part by *metB* genes, could this explain why the authors never saw reversions in the other part of the mutualistic pair, i.e. the *metA* auxotroph?
5. Can the authors expand more about the mechanisms in lines (309-313). I didn't quite get how this would work.
6. The choice of stressors (salinity and p-nitrophenol) is justified in terms of the types of cellular stress they impose. Could the authors speculate if their work can be generalized to other kinds of stress relevant to mutualistic communities in natural environments? For example, how would these mutualisms respond to nutrient limitation or temperature changes?

(Remarks on code availability)

Version 1:

Reviewer comments:

Reviewer #1

(Remarks to the Author)

In general, the authors adequately addressed the major concerns raised in my previous evaluation.

In the response to reviews (R1), an argument is made that populations may be less adaptive because they have a lower mutation rate (supply). Since mutations are mostly deleterious, I'm not sure this is too compelling, but I don't see this argument being made in the main text.

In the response to reviews (R2) and lines 291-299, the authors describe experiments using strains with naturally occurring single mutations in metabolic genes. It's unclear whether these strains have mutations in other locations throughout the genome. Presumably they do. If so, how does this influence the inferences being made given the possibility of those mutations affecting fitness directly or epistatically. This should be made clearer.

The explanation for breakdown is that it's just too hard to invest in mutualism while also contending with stress. I guess this may be the case, and perhaps the data from the laboratory experiment support this idea. But I wonder whether this explains mutualisms in nature. The authors provide some nice examples that might support their premise, but stress is pretty common

in nature. If this contributes to mutualism breakdown, then why are mutualisms so prevalent in nature? In some cases, it's been argued that mutualisms (e.g., rhizosphere microbes) convey tolerance to plants experiencing various stressors. In any case, interesting questions to be further explored.

- Jay Lennon

(Remarks on code availability)

Reviewer #2

(Remarks to the Author)

Thank you to the authors for addressing all my comments.

(Remarks on code availability)

REVIEWER COMMENTS

Reviewer #1 (Remarks to the Author):

Overview:

This paper explores how an obligate mutualism consisting of two engineered *E. coli* strains respond to environmental stress. One strain is auxotrophic for the amino acid methionine (ΔmetA) and the other strain is auxotrophic for the amino acid isoleucine (ΔilvA). These strains were passaged in 96 well plates for the equivalent 80 generations under three conditions: no stress, salinity stress, or p-nitrophenol stress. Although a few populations went extinct, most survived. When exposed to stress, this survival always involved the loss of the ΔmetA . The persistence of ΔilvA was attributed to a “bypassing of the auxotrophy”. Based on whole genome sequencing of isolates, different mutations arose under different stress treatments, which may have allowed ΔilvA to revert to an autonomous lifestyle. These findings are discussed in the context of evolutionary rescue and address questions about mutualism breakdown. The paper is reasonably well written. The figures are clear and to the point.

Thanks for this clear summary of our experimental design and main findings.

Major concerns:

Q1 It is curious that, from a demographic and evolutionary perspective, the mutualism seems very asymmetric. The ΔmetA strain is always lost, but we don't really learn much about why that's the case. Is the reason for the loss of ΔmetA due to its critical role in initiating protein synthesis, its involvement in key metabolic pathways, or its contribution to the bacterial stress response and adaptation? From Fig. 2A, it is difficult to assess the dynamics of the individual strains. It's possible that ΔmetA rapidly goes to extinction, perhaps in just a transfer or two. Might this mean that the outcomes were inevitable with the ΔmetA - ΔilvA pairing? Was the deck stacked against ΔmetA from the very start? If so, perhaps the findings are not very generalizable or as interesting as the paper suggests?

R1. We thank the reviewer for raising this interesting point. While it is possible that the ΔmetA strain might have rapidly gone to extinction from increased stress sensitivity, two lines of evidence argue against this explanation, now included in the manuscript. First, inspired by this reviewer's Q3, we conducted monoculture experiments under stress with amino acid supplementation. As shown in the **new Figure S2**, the ΔmetA strain transitions from a benign to a stressful environment without significant changes in population numbers, indicating no increased stress sensitivity. Second, the equilibrium strain ratio under benign conditions is 2.5:1 in favor of ΔilvA (**new Table S4**), suggesting a straightforward explanation: ΔmetA goes to extinction because it is disadvantaged by a smaller effective mutation supply. As the reviewer hints, methionine's critical role in initiating protein synthesis could limit the auxotroph's growth when methionine supply is scarce. This reduced population size, and hence a reduced capacity to generate adaptive mutations, may alone explain much of ΔmetA 's higher extinction rate.

Beyond the specific case at hand, this comment also led us to recognize a broader implication of our work: mutualisms are most likely inherently asymmetric. Strains (let alone species) engaged in mutualism are unlikely to have identical equilibrium ratios,

mutation rates, or stress sensitivities; and these asymmetries are probably a major destabilizing factor in nature. We now highlight the importance of asymmetries, which we believe makes the paper both more generalizable and conceptually richer than before. Here is what we have now added to the manuscript (lines 339-352):

“Notably, in all recovered populations, it was always the ΔI strain that avoided extinction. We found no evidence that the ΔM strain has a heightened stress sensitivity that may lead to rapid extinction before generating adaptive mutations (Figure S2). Instead, we found that ΔI outnumbers ΔM at equilibrium under benign conditions by a ratio of 2.5:1 (Table S4), likely explaining its greater capacity to evolve and recover. This asymmetry may stem from methionine’s critical role in initiating protein synthesis, which could limit the ΔM auxotroph’s growth when methionine is scarce. Moreover, besides population size, we cannot rule out that ΔI has a larger mutational target size for compensatory mutations compared to $\Delta M^{\Delta 2}$, as suggested by the diversity of recovered mutations and loci linked to amino acid biosynthesis (Tables S2–S3). More broadly, these observations illustrate that mutualisms are most likely inherently asymmetric: strains (let alone species) engaged in mutualism are unlikely to have identical equilibrium ratios, mutation rates, or stress sensitivities; and these asymmetries are probably a major destabilizing factor in nature that warrant future theoretical and empirical investigation.”

Q2. The paper provides three explanations for how $\Delta ilvA$ may have evolved autonomy during the experiment. These seem like plausible and testable hypotheses. Given the tractability of the model organism (*E. coli*) it’s not totally unreasonable to expect that some of these mutations could have been introduced into “clean” auxotrophs to see if they could recapitulate the phenotype of the evolved mutant. Such findings would support the assertion that these mutations are in fact compensatory, and would allow for deeper insight into the mechanisms of defection.

R2. Thank you for this very interesting suggestion. Spurred by it, we performed a complementary experiment to test whether these mutations are indeed compensatory. We leveraged the fact that many salinity- and PNP-stress-derived populations carry single mutations in metabolic genes. We grew individual colonies from these populations in M9 without isoleucine and we found that different mutations in the same gene can result in varying degrees of isoleucine auxotrophy compensation. For example, a mutation in *metB* (T322M) almost fully restored growth, while the N15S mutation showed less compensation (Figure S9). In the case of *leuA* (PNP-derived), we isolate colonies with two mutations in the same gene. We found that these mutations caused a small degree of compensation for auxotrophy and lower compared to mutations in *metB* (Figure S9). These results indeed enrich our understanding of the nuances underlying this phenomenon of reversion to autonomy. We add this new results in lines 291-299:

“To confirm that mutations in metabolic genes were responsible for the bypass of isoleucine auxotrophy. We leveraged the fact that many salinity- and PNP-stress-derived populations carry single mutations in metabolic genes. We thus isolated colonies from these populations and sequenced them to verify they carry only the mutations identified at the population level (Figure 5). We found that mutations in different genes, as well as different mutations within the same gene, resulted in varying levels of compensation for isoleucine auxotrophy (Figure S9). For instance, the T322M

mutation in metB almost fully restored growth in the absence of amino acids to the levels of the prototrophic ancestor, while the N15S mutation in metB provided weaker compensation. We also found that mutations in leuA resulted in a lower degree of compensation compared to mutations in metB.”

Figure S9. Mutations in metabolic genes can bypass isoleucine auxotrophy and restore growth to varying degrees in the absence of amino acids. a) Growth curves of different isolates in media without amino acid supplementation and without stress. In each panel, three different isolates from an independently derived population are shown, together with the prototrophic strain (grey), and the ΔI ancestor (blue). The data are presented as the mean \pm SD of 3 replicates per strain b) Loci with mutations present in at least two replicate populations or occurring at a frequency > 0.3 and c) Loci with mutations present at isolate level. Colors correspond to each isolate.

Q3. Some additional experiments came to mind that might assist with understanding the population dynamics. It might be interesting to see if similar mutations would arise if $\Delta ilvA$ strains were grown over 20 transfers in media supplemented with isoleucine and exposed to stress. If $\Delta metA$ strains were grown with supplemented methionine or mutualist-derived media and grown under the same conditions, would they still go extinct? If findings from single-strain experiments are different from cocultures, then I believe that would make a strong argument for the mutualism affecting evolutionary rescue, as stated in the introduction.

R3. Following the reviewer's suggestion, we performed a complementary evolutionary experiment where both auxotrophic strains were grown in mutualism without amino acids and as monocultures in media supplemented with the corresponding amino acid (100 μ M each for monocultures). The cultures were kept under non-stress conditions for three transfers, after which we added salinity to the media (3%). The monocultures did not exhibit the population size decrease observed in the mutualistic cocultures, whereas the mutualisms' population size decreased rapidly. We decided to end the experiment after 10 transfers, at which point most mutualisms had recovered (80%). We also sequenced five populations from both the I and M monocultures at the end of this experiment and did not find any of the mutations in metabolic genes that were previously observed in the mutualism. These experimental results have been included in the revised manuscript (lines 147-153):

“We also explored the adaptive response of the auxotrophic strains when amino acids are supplemented in the medium and they do not rely on their partner (Figure S2, a). We found that neither monoculture exhibited the population size decrease observed in the mutualistic populations. Sequencing revealed no single mutation becoming fixed in any of these populations, and those reaching high-frequency were different from the ones observed in the main experiments with the mutualists (Figure S2, b-c). These findings further support the notion that the mutualism makes populations more prone to extinction under stress.”

Figure S2. Auxotrophic strains are less affected by stress when supplemented with amino acids. a) Population dynamics of the obligate mutualism consortium (green) and auxotrophic strains I and M (blue and yellow, respectively). The pink background indicates exposure to salinity. The experiment consisted of 96 independent populations for each treatment. The media was supplemented with 100 μ M of isoleucine or methionine, respectively. b, c) Genes with mutations present in at least two replicate populations or occurring at a frequency > 0.1 in five independent evolutionary replicates from monocultures I (blue) and M (yellow), respectively. Color shades indicate the frequency of each mutation, with darker shades indicating higher mutation frequencies. Specific mutations are detailed in Tables S5-6.

Q4. I appreciate the use of a model system to test general ideas, so perhaps this is somewhat semantic, but an ecological “community” is made up of two or more species. Similarly, mutualisms involve two or more species. In this case, the experiments involve two engineered derivatives of MG165 *E. coli*. In other words, this is very much a population, not a community.

R4. We agree with your point and, to avoid any confusion, we will adopt the term “consortium” instead of referring to our system as a “community”. This terminology aligns with previous literature, such as the study by Pauli et al. (*Nat. Commun.*, 2022), which also used the term “consortium” for a mutualistic pair of auxotrophic *E. coli* strains. By using this term, we aim to accurately describe the metabolic interdependencies in our system while maintaining consistency with established nomenclature in the field.

Other concerns:

Q5. The section titled “Mutualists can survive lower stress levels than the prototrophic strain” seems a bit out of place. The title is confusing; it reads as though the mutualists have an ability that the prototrophs do not, when the mutualists have an inability to survive the same caliber of stress as the prototroph.

R5. We agree that the current wording could be misleading. To clarify, we have revised the title to: “*Mutualists are less tolerant to high stress levels compared to the prototrophic strain.*” See line 195.

Q6. In the last sentence of the section mentioned above, it seems there is an error reading, “the auxotrophic strain is able to survive stress levels that would drive the mutualism towards extinction.” Is this supposed to be “prototrophic”?

R6. This sentence should indeed read “*the prototrophic strain is able to survive stress levels that would drive the mutualism towards extinction.*” We have corrected this in the revised manuscript. See lines 209-210.

Q7. “Toxicity assays” has a typo in line 4 (“ot” instead of “or”).

R7. We have corrected “ot” to “or” in the revised manuscript. See line 457.

Reviewer #2 (Remarks to the Author):

This manuscript by Melero-Jimenez and colleagues explores the dynamics of evolutionary rescue in a microbial community that is designed to have an obligate mutualism through the cross-feeding for amino acids. The authors provide strong evidence that more than 80% of communities avoid extinction under stress conditions, and that the primary mechanism for this the rescue is reversion to metabolic autonomy in one of the partners. These results have significant implications for the understanding of microbial community resilience under environmental stress, especially in the context of mutualistic relationships. I think the manuscript is well written, methodologically sound, and combines ecological data with genomic data provides interesting insights into the role of interactions and mutations in evolutionary rescue. The writing is very clear and I appreciate the hypothesis driven design of the work. However, I think several points need further clarification and expansion to strengthen the conclusions drawn and address potential gaps.

Thanks for your clear summary and positive appraisal of our work.

Q1. Title: I think the title could be made more specific- What is missing is that reversion of metabolic autonomy causes a breakdown in the mutualism (in most cases). The current title leads one to believe that the evolutionary rescue saved the mutualism.

R1. We have revised the title to make it clearer and more specific, focusing on how mutualism breakdown leads to evolutionary rescue in the mutualistic consortium. The updated title is *"Mutualism breakdown underpins evolutionary rescue in an obligate cross-feeding bacterial consortium"*

Q2. Discussion: I would have liked a more detailed discussion of what this work can inform one about the stability of microbial/mutualistic communities and where it could be useful. The authors allude to this in the start of the main text (lines 53-56) but never revisit this again.

R2. In response to the reviewer's comment, we have expanded the discussion to address the broader implications of our findings (lines 386-394):

"Understanding the processes that lead to the breakdown of mutualistic interactions could be useful in many fields. In agriculture, identifying when plant-microbe mutualism⁴⁶ as those between nitrogen-fixing bacteria and legumes⁵⁰ begin to break can inform soil management and the selection of resilient bacterial strains, thereby reducing crop loss^{45,51}. In medicine, insights into the thresholds at which host-microbe mutualisms^{52,53} break—whether due to antibiotics⁵⁴, dietary changes⁵⁵, or infections⁵⁶—could guide preventive therapies using targeted prebiotics or probiotics to maintain a balanced microbiome^{57,58}. In biological conservation, the capacity to anticipate mutualism breakdown allows design more effective conservation measures and politics."

Moreover, as explained in our response to Reviewer 1, we now highlight that our work suggests mutualisms are most likely inherently asymmetric: strains (let alone species) engaged in mutualism are unlikely to have identical equilibrium ratios, mutation rates, or stress sensitivities; and these asymmetries are probably a major destabilizing factor in nature that warrant future theoretical and empirical investigation. The appropriate passage is now in 348-352:

"More broadly, these observations illustrate that mutualisms are most likely inherently asymmetric: strains (let alone species) engaged in mutualism are unlikely to have identical equilibrium ratios, mutation rates, or stress sensitivities; and these asymmetries are probably a major destabilizing factor in nature that warrant future theoretical and empirical investigation."

Q3. Can the authors provide (reasonable) speculation as to what mechanisms (molecular or otherwise) that strains that constitute mutualistic interactions are sensitive to stress?

R3. We suggest that strains engaged in mutualism are particularly sensitive to stress because they face two challenges simultaneously: dealing with the stress while maintaining mutualistic interactions to acquire essential amino acids. Unlike prototrophic strains, auxotrophic strains rely entirely on metabolite exchange, which may involve an increased membrane permeability to facilitate nutrient uptake. This could make the cells more vulnerable to stressors, such as osmotic changes or toxins. Another possibility is that strains may experience a decrease in the biosynthesis of amino acids due to the reallocation of cellular resources under stress. As a result, the lack of essential amino acids impairs protein synthesis, reduces the consortia growth, and weakens the

consortium's ability to respond to environmental challenges. We have now added the following to lines 319-326:

“We speculate here that strains in mutualism are particularly sensitive to stress because they face two simultaneous challenges: coping with stress while maintaining mutualistic interactions to acquire essential amino acids. Unlike prototrophic strains, auxotrophs rely entirely on metabolite exchange, which may involve increased membrane permeability, making them more vulnerable to stressors like osmotic changes or toxins. Alternatively, stress may reduce amino acid biosynthesis due to resource reallocation, impairing protein synthesis, slowing consortium growth, and weakening its response to environmental challenges.”

Q4. Since the reversion is driven in part by *metB* genes, could this explain why the authors never saw reversions in the other part of the mutualistic pair, i.e. the *metA* auxotroph?

R4. Good question. However, we do not see a strong reason to expect that mutations bypassing a blockade in the isoleucine biosynthesis pathway would also bypass a blockade in the methionine biosynthesis pathway. As explained to Reviewer 1 (and now included in the text, lines 291–299), we favor a demographic explanation: the effective population size of ΔM is substantially smaller than that of ΔI . We now provide new data supporting that ΔI outnumbers ΔM by a ratio of 2.5:1 at the start of the experiment. This difference likely provides ΔI with a larger effective mutation supply, increasing its chances to evolve and recover compared to ΔM .

That said, and to the reviewer’s point, we acknowledge that other factors may contribute to the observed trend, such as a larger mutational target size for bypassing the *ilvA* auxotrophy compared to *metB*. Future work exploring the underlying metabolic intricacies of these mutants could help provide a more definitive answer.

Q5. Can the authors expand more about the mechanisms in lines (309-313). I didn’t quite get how this would work.

R5. We have expanded about the mechanism in R3 (lines 319-326), and we also add some examples to clarify this point (lines 328-331):

“This latter possibility has been suggested in legume-rhizobia mutualisms, where bacteria often stop fixing nitrogen in nitrogen-rich soils, conserving energy that would otherwise support the plant and redirecting it toward their own growth⁴². Similarly, in coral-algae mutualism, algae may reduce the nutrients they share with their coral hosts when nutrients are abundant⁴³.”

Q6. The choice of stressors (salinity and p-nitrophenol) is justified in terms of the types of cellular stress they impose. Could the authors speculate if their work can be generalized to other kinds of stress relevant to mutualistic communities in natural environments? For example, how would these mutualisms respond to nutrient limitation or temperature changes?

R6. We speculate in the discussion about how this work can be generalized to other mutualism in natural environments (lines 366-384):

“Here, we selected stressors that induce specific cellular stresses, challenging the survival of both partners. However, the breakdown of mutualism observed here may also occur in other mutualisms and environmental stresses. Our results are particularly relevant for understanding when breakdowns can occur in other bacterial mutualisms, such as cross-protection or nutrient exchange. The breakdown may also happen in other environments where engaging in a mutualism constrains the adaptation of both partners, thus creating a situation in which becoming autonomous may be the most effective strategy for facing stress. For example, in nutrient-limiting conditions, the growth of both partners would be reduced, and they would already find it hard to exchange those amino acids, which likely ends in the extinction or breakdown of their mutualistic interaction. This scenario mirrors what happens in nature in fungi-plant mutualisms⁴⁵. Under nutrient deficiencies, plants cannot provide sufficient amount of carbohydrates to fungi; they further weaken the mutualism and generally lead to declines in health of mutualism^{46,47}. It has been also observed that nitrogen-fixing bacteria disrupt the mutualism between legume and rhizobia under nutrient-poor soils^{42,48}. Other stresses, such as thermal stress, can disrupt metabolic processes essential for amino acid exchange by impairing enzyme activity and membrane stability, which can lead to reduced nutrient transport and potential mutualism breakdown. We hypothesize that thermal stress will hinder the transport of amino acids in our system in a similar way that increases in ocean temperature disrupt nutrient exchange between coral and their symbiotic algae to cause coral bleaching⁴⁹.”

REVIEWERS' COMMENTS

Reviewer #1 (Remarks to the Author):

In general, the authors adequately addressed the major concerns raised in my previous evaluation.

Q1. In the response to reviews (R1), an argument is made that populations may be less adaptive because they have a lower mutation rate (supply). Since mutations are mostly deleterious, I'm not sure this is too compelling, but I don't see this argument being made in the main text.

R1. Thank you for this feedback. This is an important, and perhaps subtle point: rather than arguing that populations are less adaptive due to a lower mutation rate, we suggest that the higher extinction rate of the $\Delta metA$ strain may be explained by the fact that they have a lower population size. We argue that because there is an equilibrium strain ratio of 2.5:1 in favor of $\Delta ilvA$ under non-stress conditions (Table S4), $\Delta metA$ has a smaller effective population size and lower mutation supply (lower possibilities to find a revertant mutant). This demographic disadvantage likely increasing $\Delta metA$'s likelihood of extinction.

We clarified this point in the main text lines **309-312**:

“Instead, we found that ΔI outnumbers ΔM at equilibrium under benign conditions by a ratio of 2.5:1 (Supplementary Table 4), this difference likely provides ΔI with a larger effective mutation supply, increasing its chances to evolve and recover compared to ΔM .”

Q2. In the response to reviews (R2) and lines 291-299, the authors describe experiments using strains with naturally occurring single mutations in metabolic genes. It's unclear whether these strains have mutations in other locations throughout the genome. Presumably they do. If so, how does this influence the inferences being made given the possibility of those mutations affecting fitness directly or epistatically. This should be made clearer.

R2. As noted in our previous rebuttal letter, we isolated clones from populations that had only single mutations in metabolic genes as revealed by whole-population sequencing data. We then sequenced these newly isolated clones to confirm that no additional mutations were present in their genomes. This way, we can be certain that our interpretations are based solely on the metabolic mutations, with no other mutations affecting fitness or acting epistatically (see Supplementary Fig. 9). We describe this in the main text (lines **268–272**), where we emphasize that these are the only mutations identified in the genomes of the strains we phenotyped in this section.

Q3. The explanation for breakdown is that it's just too hard to invest in mutualism while also contending with stress. I guess this may be the case, and perhaps the data from the laboratory experiment support this idea. But I wonder whether this explains mutualisms

in nature. The authors provide some nice examples that might support their premise, but stress is pretty common in nature. If this contributes to mutualism breakdown, then why are mutualisms so prevalent in nature? In some cases, it's been argued that mutualisms (e.g., rhizosphere microbes) convey tolerance to plants experiencing various stressors. In any case, interesting questions to be further explored.

R3. Thanks for this interesting point. Our findings show that stress can destabilize mutualisms when the costs of maintaining them become too high. However, this does not exclude the existence of stress-resistant mutualisms. As the reviewer points out, certain mutualisms, such as those involving rhizosphere microbes, can enhance stress tolerance. Moreover, previous studies have argued that the generation of mutualism may be likely under stable conditions (e.g. the Black Queen Hypothesis). So, the prevalence of mutualisms in nature may be shaped by a balance between their generation and disruption and vary depending on environmental and genetic factors. We whole heartily agree that understanding the conditions under which stress weakens or strengthens mutualistic interactions is an important and fascinating direction for future research. We add this point in lines 370-372:

“If stress is conducive to mutualism breakdown, how come mutualisms are so prevalent in nature? Future work in other model systems will help elucidate the generality of our results.”

Reviewer #2 (Remarks to the Author):

Q1. Thank you to the authors for addressing all my comments.

R1. Thank you for your thoughtful feedback.